# CYK Parsing over Distributed Representations

**Fabio Massimo Zanzotto** [1,*] , **Giorgio Satta** [2] **and Giordano Cristini** [1,†]

1   Department of Enterprise Engineering, University of Rome Tor Vergata, Viale del Politecnico 1,
    00133 Roma, Italy; giordano.cristini@gmail.com or giordano.cristini@exprivia.com
2   Department of Information Engineering, University of Padua, via Gradenigo 6/A, 35131 Padova, Italy;
    satta@dei.unipd.it
*   Correspondence: fabio.massimo.zanzotto@uniroma2.it
†   Current address: Exprivia SpA, Viale del Tintoretto 432, 00142 Roma, Italy.

**Abstract:** Parsing is a key task in computer science, with applications in compilers, natural language processing, syntactic pattern matching, and formal language theory. With the recent development of deep learning techniques, several artificial intelligence applications, especially in natural language processing, have combined traditional parsing methods with neural networks to drive the search in the parsing space, resulting in hybrid architectures using both symbolic and distributed representations. In this article, we show that existing symbolic parsing algorithms for context-free languages can cross the border and be entirely formulated over distributed representations. To this end, we introduce a version of the traditional Cocke–Younger–Kasami (CYK) algorithm, called distributed (D)-CYK, which is entirely defined over distributed representations. D-CYK uses matrix multiplication on real number matrices of a size independent of the length of the input string. These operations are compatible with recurrent neural networks. Preliminary experiments show that D-CYK approximates the original CYK algorithm. By showing that CYK can be entirely performed on distributed representations, we open the way to the definition of recurrent layer neural networks that can process general context-free languages.

**Keywords:** parsing algorithms; neural networks; distributed representations; formal languages

## 1. Introduction

In computer science, parsing is defined as the process of identifying one or more syntactic structures for an input string, according to some underlying formal grammar specifying the syntax of the reference language. Most important applications where parsing is exploited are compilers for programming languages, natural language processing, markup languages, syntactic pattern matching, and data structures in general. In the large majority of cases, the underlying grammar is a context-free grammar (CFG), and the syntactic structure is represented as a parse tree; for definitions, see for instance [1], Chapter 5.

Parsing algorithms based on CFGs have a long tradition in computer science, starting as early as in 1960s. These algorithms can be grouped into two broad classes, depending on whether they can deal with ambiguous grammars or not, where a grammar is called ambiguous if it can assign more than one parse tree to some of its generated strings. For non-ambiguous grammars, specialized techniques such as LLand LR [2] can efficiently parse input in linear time. These algorithms are typically exploited within compilers for programming languages [3]. On the other hand, general parsing methods have been developed for ambiguous CFGs that construct compact representations, called parse forests, of all possible parse trees assigned by the underlying grammar to the input string. These algorithms use dynamic programming techniques and runin polynomial time in the length of the input string;

see for instance [4] for an overview. Parsing algorithms for ambiguous CFGs are widespread in areas such as natural language processing, where different parse trees are associated with different semantic interpretations [5].

The Cocke–Younger–Kasami algorithm (CYK) [6–8] was the first parsing method for ambiguous CFGs to be discovered. This algorithm is at the core of many common algorithms for natural language parsing, where it is used in combination with probabilistic methods and other machine learning techniques to estimate the probabilities for grammar rules based on large datasets [9], as well as to produce a parse forest of all parse trees for an input string and retrieve the trees having the overall highest probability [10].

With the recent development of deep learning techniques in artificial intelligence [11], natural language parsing has switched to so-called distributed representations, in which linguistic information is encoded into real number vectors and processing is carried out by means of matrix multiplication operations. Distributed vectors can capture a large number of syntactic and semantic information, allowing linguistic meaningful generalizations and resulting in improved parsing accuracy. Neural network parsers have then been developed, running a neural network on the top of some parsing algorithm for ambiguous CFGs, as the CYK algorithm introduced above. A more detailed discussion of this approach is reported in Section 2.

As described above, neural network parsers based on CFGs are hybrid architectures, where a neural network using distributed representations is exploited to drive the search in a space defined according to some symbolic grammar and some discrete data structure, as for instance the parsing table used by the already mentioned CYK algorithm. In these architectures, the high-level, human-readable, symbolic representation is therefore kept separate from the low-level distributed representation. In this article, we attempt to show that traditional parsing algorithms can cross the border of distributed representations and can be entirely reformulated in terms of matrix multiplication operations. More precisely, the contributions of our work can be stated as follows:

- We propose a distributed version of the CYK algorithm, called D-CYK, that works solely on the basis of distributed representations and matrix multiplication;
- We show how to implement D-CYK by means of a standard recurrent neural network.

We also run some preliminary experiments on artificial CFGs, showing that D-CYK is effective at mimicking CYK for small size grammars and strings. Our work therefore opens the way to the design of parsing algorithms that are entirely based on neural networks, without any auxiliary symbolic data structures.

More technically, our main result is achieved by transforming the parsing table at the base of the CYK algorithm, introduced in Section 3.1, into two square matrices of fixed size, defined over real numbers, in such a way that the basic operations of the CYK algorithm can be implemented using matrix multiplications. Implementations of the CYK algorithm using matrix multiplications are well known in the literature [12,13], but they use symbolic representations, while in our proposal, grammar symbols, as well as constituent indices are all encoded into real numbers and in a distributed way. A second, important difference is that in the standard CYK algorithm, the parsing table has size $(n + 1) \times (n + 1)$, with $n$ the length of the input string, while our D-CYK manipulates matrices of size $d \times d$, where $d$ depends on the distributed representation only. This means that, to some extent, we can parse input strings of different lengths without changing the matrix size in the D-CYK algorithm. We are not aware of any parsing algorithm for CFGs having such a property. This is the crucial property that allows us to implement the D-CYK algorithm using recurrent neural networks, without the addition of any symbolic data structure.

## 2. Related Work

Early attempts to formulate context-free parsing within the framework of neural networks can be found in [14,15]. Fanty [14] proposed a feed-forward network whose connections reflect the structure

of the rules in the CFG and whose activations directly encode the content of the CYK table, introduced in Section 3.1, viewed as an and-or graph. The number of parameters of the network therefore depends on the length of the input string. Inhibition connections in combination with ranking are used to resolve disambiguation. Nijholt [15] generalized this approach to CFGs that are not in Chomsky normal form, by proposing a network that mimics the content of the parse table as constructed by the Earleyalgorithm [16], a second standard parsing method based on CFGs. In both of these early approaches, the grammar is represented by the topology of the network, and there is no use for distributed representation for encoding the rules of the grammar or parsing information.

Natural language processing is perhaps the area where most research has been carried out in order to develop neural network parsers based on general CFGs, following the recent development of deep learning techniques. These approaches can be roughly divided into two classes: approaches that use a neural network to drive a push-down automaton implementing the CFG of interest and approaches that enrich a CFG with a neural network and then parse using a CYK-like algorithm. Approaches based on a push-down automaton include Henderson [17], which is the first successful demonstration of the use of neural networks in large-scale parsing experiments with CFGs. This work exploits a specialization of recurrent neural networks, called simple synchrony networks, to compute a probability for each step in a push-down automaton implementing the CFG of interest. Each action of the automaton is conditioned on all previous parsing steps, without making any specific hard-wired independence assumption. In this way, the parser is able to compute a representation of the entire derivation history. The parser is then run in combination with beam search and other pruning strategies. Similarly, Chen and Manning [18] ran a push-down automaton and used a feed-forward neural network to drive a greedy search through the parsing space. This results in a fast and compact parser; however, the parser is not able to represent the entire parsing history. Watanabe and Sumita [19] used a shift-reduce parser for CFGs with binary and unary nodes, combined with beam search. The parser employs a recurrent neural network to compute a distributed representation for the parsing history based on unlimited portions of the stack and the input queue. Dyer et al. [20] extended long short-term memory networks (a specialization of recurrent neural networks) to a stack data structure and used these to drive the actions of a push-down automaton in a greedy mode. The neural network then provides a distributed representation of all of the stack content, the entire portion of the string still to be read, and all of the parsing actions taken so far.

Approaches based on a generative grammar include Socher et al. [21], who introduced compositional vector grammars that combine a probabilistic CFG and a recurrent neural network. The formalism uses a dual representation of nodes as discrete categories and real-valued vectors, where the latter are computed by applying the neural network weights on the basis of the specific syntactic categories at hand. In this way, each constituent is associated with a distributed representation potentially encoding its entire derivation history. The parsing algorithm is based on the CYK method combined with a beam search strategy and applies the neural network component in a second pass to rerank the obtained derivations. Dyer et al. [22] introduced a formalism called recurrent neural network grammars, which is a probabilistic generative model based on CFGs, in which rewriting is parameterized using recurrent neural networks that condition on the entire syntactic derivation history. The authors introduced learning algorithms for recurrent neural network grammars and developed a parser incorporating top-down information.

All of the above approaches use a mix of symbolic methods and distributed representations obtained by means of neural networks. One notable exception to this trend can be found in [23], where parsing was viewed as a sequence-to-sequence problem. The authors represented parse trees in a linearized form and exploited a long short-term memory network, which first encodes the input string $w$ into an internal representation $h_w$ and then uses an attention mechanism over $w$ to decode $h_w$ into a linearized tree for $w$. This approach is entirely based on distributed representation and matrix multiplication; no auxiliary symbolic data structure is used. Despite the fact that Vinyals et al. [23] achieved state-of-the-art performance in natural language parsing, there is still a theoretical issue with the use of recurrent neural networks for context-free language parsing. Weiss et al. [24] and Suzgun et al. [25] showed that a long

short-term memory network can process languages structurally richer than regular languages, but cannot achieve the full generative power of the class of context-free languages.

In contrast with the hybrid architectures presented above, our proposal is entirely based on a neural network architecture and on a distributed representation, entirely giving up the use of auxiliary discrete data structures and symbolic representations. Furthermore, in contrast with the sequence-to-sequence approach, our proposal is a faithful implementation of the CYK algorithm and can in principle achieve the full generative power of the class of context-free languages.

## 3. Preliminaries

In this section, we introduce the basics about the CYK algorithm and overview a class of distributed representations called holographic reduced representation.

### 3.1. CYK Algorithm

The CYK algorithm is a classical algorithm for recognition/parsing based on context-free grammars (CFGs), using dynamic programming. We provide here a brief description of the algorithm in order to introduce the notation used in later sections; we closely follow the presentation in [4], and we assume the reader is familiar with the formalism of CFG ([1], Chapter 5). The algorithm requires CFGs in Chomsky normal form, where each rule has the form $A \to BC$ or $A \to a$, where $A$, $B$, and $C$ are nonterminal symbols and $a$ is a terminal symbol. We write $R$ to denote the set of all rules of the grammar and $NT$ to denote the set of all its nonterminals.

Given an input string $w = a_1 a_2 \cdots a_n$, $n \geq 1$, where each $a_i$ is an alphabet symbol, the algorithm uses a two-dimensional table $P$ of size $(n+1) \times (n+1)$, where each entry stores a set of nonterminals representing partial parses of the input string. More precisely, for $0 \leq i < j \leq n$, a nonterminal $A$ belongs to set $P[i, j]$ if and only if there exists a parse tree with root $A$ generating the substring $a_{i+1} \cdots a_j$ of $w$. Thus, $w$ can be parsed if the initial nonterminal of the grammar $S$ is added to $P[0, n]$. Algorithm 1 shows how table $P$ is populated. $P$ is first initialized using unary rules, at Line 3. Then, each entry $P[i, j]$ is filled at Line 11 by looking at pairs $P[i, k]$ and $P[k, j]$ and by using binary rules.

---

**Algorithm 1** CYK(string $w = a_1 a_2 \cdots a_n$, rule set $R$) **return** table $P$.

---

 1: **for** $i \leftarrow 1$ to $n$ **do**
 2:     **for** each $A \to a_i$ in $R$ **do**
 3:         add $A$ to $P[i-1, i]$
 4:     **end for**
 5: **end for**
 6: **for** $j \leftarrow 2$ to $n$ **do**
 7:     **for** $i \leftarrow j-2$ to $0$ **do**
 8:         **for** $k \leftarrow i+1$ to $j-1$ **do**
 9:             **for** each $A \to BC$ in $R$ **do**
10:                 **if** $B \in P[i, k]$ and $C \in P[k, j]$ **then**
11:                     add $A$ to $P[i, j]$
12:                 **end if**
13:             **end for**
14:         **end for**
15:     **end for**
16: **end for**

---

A running example is presented in Figure 1, showing a set $R$ of grammar rules along with the table $P$ produced by the algorithm when processing the input string $w = aab$ (the right part of this figure should be ignored by now). For instance, $S$ is added to $P[1, 3]$, since $D \in P[1, 2]$, $E \in P[2, 3]$, and $(S \to DE) \in R$. Since $S \in P[0, 3]$, we conclude that $w$ can be generated by the grammar.

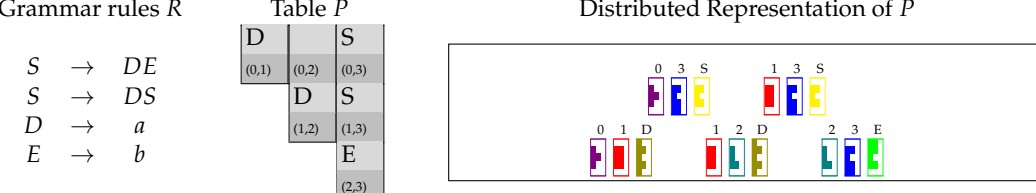

**Figure 1.** A simple context-free grammar (CFG), the Cocke–Younger–Kasami (CYK) parsing table $P$ for the input string $w = aab$ as constructed by Algorithm 1, and the "distributed" representation $\mathbf{P}_{left}$ of $P$ in Tetris-like notation.

### 3.2. Distributed Representations with Holographic Reduced Representations

Holographic reduced representations (HRRs), introduced in [26] and extended in [27], are distributed representations well suited for our aim of encoding the two-dimensional parsing table $P$ of the CYK algorithm and for implementing the operation of selecting the content of its cells $P[i, j]$. In the following, we introduce the operations we use, along with a graphical way to represent their properties. The graphical representation is based on Tetris-like pieces.

These HRRs represent sequences of symbols $s = s_1 \ldots s_n$ in vectors $\vec{s}$ by composing vectors $\vec{s_i}$ of symbols $s_i$ in the sequences. Hence, these representations offer an encoder and a decoder for sequences in vectors. The encoder and the decoder are not learned, but rely on the statistical properties of random vectors and the properties of a basic operation for composing vectors. The basic operation is circular convolution [26], extended to shuffled circular convolution in [27], which is non-commutative and, then, alleviates the problem of confusing sequences with the same symbols in different order.

The starting point of a distributed representation and, hence, of HRR is how to encode symbols into vectors: symbol $a$ is encoded using a random vector $\vec{a} \in \mathbb{R}^d$ drawn from a multivariate normal distribution $\vec{a} \sim N(0, \mathbf{I}\frac{1}{\sqrt{d}})$. These are used as basis vectors for the Johnson–Lindenstrauss transform [28], as well as for random indexing [29]. The main property of these random vectors is the following:

$$\vec{a}^\top \vec{b} \quad \approx \quad \begin{cases} 1 & \text{if } \vec{a} = \vec{b} \\ 0 & \text{if } \vec{a} \neq \vec{b} \end{cases}$$

In this paper, we use the matrix representation of the shuffled circular convolution introduced in [27]. In this way, symbols are represented in a way where composition is just matrix multiplication and the inverse operation is matrix transposition. Given the above symbol encoding, we can define a basic operation $[\,]^\oplus$ and its approximate inverse $[\,]^\ominus$. These operations take as input a symbol and provide a matrix in $\mathbb{R}^{d \times d}$ and are the basis for our encoding and decoding. The first operation is defined as:

$$[a]^\oplus \quad = \quad \mathrm{A}_\circ \Phi \,,$$

where $\Phi$ is a permutation matrix to obtain the shuffling [27] and $\mathrm{A}_\circ$ is the circulant matrix of the vector $\vec{a}^\top = \begin{pmatrix} a_0 & a_1 & \ldots & a_{d-1} \end{pmatrix}$, that is:

$$\mathrm{A}_\circ = \begin{bmatrix} a_0 & a_{d-1} & \ldots & a_1 \\ a_1 & a_0 & \ldots & a_2 \\ \vdots & \ddots & & \vdots \\ a_{d-2} & a_{d-3} & \ldots & a_{d-1} \\ a_{d-1} & a_{d-2} & \ldots & a_0 \end{bmatrix} = \begin{bmatrix} | & | & \ldots & | \\ s_0(\vec{a}) & s_1(\vec{a}) & \ldots & s_{d-1}(\vec{a}) \\ | & | & \ldots & | \end{bmatrix}$$

while $s_i(\vec{a})$ is the circular shifting of $i$ positions of the vector $\vec{a}$. Circulant matrices are used to describe circular convolution. In fact, $\vec{a} * \vec{b} = A_\circ \vec{b} = B_\circ \vec{a}$ where $*$ is circular convolution. This operation has a nice approximated inverse in:

$$[a]^\ominus \;\;=\;\; \Phi^\top A_\circ^\top .$$

We then have:

$$[a]^\oplus[b]^\ominus \;\;\approx\;\; \begin{cases} \mathbf{I} & \text{if } \vec{a} = \vec{b} \\ \mathbf{0} & \text{if } \vec{a} \neq \vec{b} \end{cases}$$

since $\Phi$ is a permutation matrix and therefore $\Phi\Phi^\top = I$ and since:

$$A_\circ^\top B_\circ \;\;\approx\;\; \begin{cases} \mathbf{I} & \text{if } \vec{a} = \vec{b} \\ \mathbf{0} & \text{if } \vec{a} \neq \vec{b} \end{cases}$$

due to the fact that $A_\circ$ and $B_\circ$ are circulant matrices based on random vectors $\vec{a}, \vec{b} \sim N(0, \mathbf{I}\frac{1}{\sqrt{d}})$; hence, $s_i(\vec{a})^\top s_j(\vec{b}) \approx 1$ if both $i = j$ and $\vec{a} = \vec{b}$, and $s_i(\vec{a})^\top s_j(\vec{b}) \approx 0$ otherwise. Finally, the permutation matrix $\Phi$ is used to enforce non-commutativity in the matrix product such as $[a]^\oplus[b]^\oplus[c]^\oplus$.

With the $[\;]^\oplus$ and $[\;]^\ominus$ operations at hand, we can now encode and decode strings, that is finite sequences of symbols. As an example, the string *abc* can be represented as the matrix product $[a]^\oplus[b]^\oplus[c]^\oplus$. In fact, we can check that $[a]^\oplus[b]^\oplus[c]^\oplus$ starts with *a*, but not with *b* or with *c*, since we have $[a]^\ominus[a]^\oplus[b]^\oplus[c]^\oplus \approx [b]^\oplus[c]^\oplus$, which is different from $\mathbf{0}$, while $[b]^\ominus[a]^\oplus[b]^\oplus[c]^\oplus \approx \mathbf{0}$ and $[c]^\ominus[a]^\oplus[b]^\oplus[c]^\oplus \approx \mathbf{0}$. Knowing that $[a]^\oplus[b]^\oplus[c]^\oplus$ starts with *a*, we can also check that the second symbol in $[a]^\oplus[b]^\oplus[c]^\oplus$ is *b*, since $[b]^\ominus[a]^\ominus[a]^\oplus[b]^\oplus[c]^\oplus$ is different from $\mathbf{0}$. Finally, knowing that $[a]^\oplus[b]^\oplus[c]^\oplus$ starts with *ab*, we can check that the string ends in *c*, since $[c]^\ominus[b]^\ominus[a]^\ominus[a]^\oplus[b]^\oplus[c]^\oplus \approx \mathbf{I}$.

Using the above operations, we can also encode sets of strings. For instance, the string set $\mathcal{S} = \{abS, DSa\}$ is represented as the sum of matrix products $[a]^\oplus[b]^\oplus[S]^\oplus + [D]^\oplus[S]^\oplus[a]^\oplus$. We can then test whether $abS \in \mathcal{S}$ by computing the matrix product $[S]^\ominus[b]^\ominus[a]^\ominus([a]^\oplus[b]^\oplus[S]^\oplus + [D]^\oplus[S]^\oplus[a]^\oplus) \approx \mathbf{I}$, meaning that the answer is positive. Similarly, $aDS \in \mathcal{S}$ is false, since $[S]^\ominus[D]^\ominus[a]^\ominus([a]^\oplus[b]^\oplus[S]^\oplus + [D]^\oplus[S]^\oplus[a]^\oplus) \approx \mathbf{0}$. We can also test whether there is any string in $\mathcal{S}$ starting with *a*, by computing $[a]^\ominus([a]^\oplus[b]^\oplus[S]^\oplus + [D]^\oplus[S]^\oplus[a]^\oplus) \approx [b]^\oplus[S]^\oplus$ and providing a positive answer since the result is different from $\mathbf{0}$.

Not only can our operations be used to encode sets, as described above; they can also be used to encode multiple sets, that is they can keep a count of the number of occurrences of a given symbol/string within a collection. For instance, consider the multi-set consisting of two occurrences of symbol *a*. This can be encoded by means of the sum $[a]^\oplus + [a]^\oplus$. In fact, we can test the number of occurrences of symbol *a* in the multi-set using the product $[a]^\ominus([a]^\oplus + [a]^\oplus) \approx \mathbf{I} + \mathbf{I} = 2\mathbf{I}$.

Our operations have the nice property of deleting symbols in a chain of matrix multiplication if the opposed symbols are contiguous. In fact, two contiguous opposed symbols become the identity matrix, which is invariant with respect to matrix multiplication. This behavior seems to be similar to what happens for the Tetris game where pieces delete lines when shapes complement holes in lines. Hence, to visualize the encoding and decoding implemented by the above operations, we will use a graphical representation based on Tetris. Symbols under the above operations are represented as Tetris pieces: for example, $[a]^\oplus = $ , $[a]^\ominus = $ , $[b]^\oplus = $ , and $[b]^\ominus = $ . In this way strings are sequences of pieces; for example,  encodes *abS* (equivalently, $[a]^\oplus[b]^\oplus[S]^\oplus$). Then, like in Tetris, elements with complementary shapes are canceled out and removed from a sequence; for example, if  is applied

to the left of [image], the result is [image] as [image] disappears. Sets of strings (sums of matrix products) are represented in boxes, as for instance:

$$L \quad = \quad \boxed{\text{[image]} \qquad \text{[image]}}$$

which encodes set $\{abS, DSa\}$ (equivalently, $[a]^{\oplus}[b]^{\oplus}[S]^{\oplus} + [D]^{\oplus}[S]^{\oplus}[a]^{\oplus}$). In addition to the usual Tetris rules, we assume here that an element with a certain shape will select from a box only elements with the complementary shape facing it. For instance, if [image] is applied to the left of the above box $L$, the result is the new box:

$$L' \quad = \quad \boxed{\text{[image]}}$$

as [image] selects [image] but not [image].

With the encoding introduced above and with the Tetris metaphor, we can describe our model to encode $P$ tables as matrices of real numbers, and we can implement CFG rule applications by means of matrix multiplication, as discussed in the next section.

## 4. The CYK Algorithm on Distributed Representations

The distributed CYK algorithm (D-CYK) is our version of the CYK algorithm running over distributed representations and using matrix algebra. As the traditional CYK, our algorithm recognizes whether or not a string $w$ can be generated by a CFG with a set of rules $R$ in Chomsky normal form. Yet, unlike the traditional CYK algorithm, the parsing table $P$ and the rule set $R$ are encoded through matrices in $\mathbb{R}^{d \times d}$, using the distributed representation of Section 3.2, and rule application is obtained with matrix algebra. Here, $d$ is some fixed natural number independent of the length of $w$.

In this section, we describe how the D-CYK algorithm encodes:

(i)     the table $P$ by means of two matrices $\mathbf{P}_{left}$ and $\mathbf{P}_{right}$;
(ii)    the unary rules in $R$ by means of a matrix $\mathbf{R}_u[A]$ for each nonterminal symbol $A \in NT$; and
(iii)   the binary rules in $R$ by means of matrices $\mathbf{R}_b[A]$ for each $A \in NT$.

We also specify the steps of the D-CYK algorithm and illustrate its execution using the running example of Figure 1.

### 4.1. Encoding the Table P in matrices $\mathbf{P}_{left}$ and $\mathbf{P}_{right}$

The table $P$ of the CYK algorithm can be seen as a collection of triples $(i, j, A)$, where $A$ is a nonterminal symbol from the grammar. More precisely, the collection of triples contains element $(i, j, A)$ if and only if $A \in P[i, j]$. Given the representation of Section 3.2, the table $P$ is then encoded by means of two matrices $\mathbf{P}_{left}$ and $\mathbf{P}_{right}$ in $\mathbb{R}^{d \times d}$. The reason why we need two different matrices to encode $P$ will become apparent later on, in Section 4.3. Matrices $\mathbf{P}_{left}$ and $\mathbf{P}_{right}$ contain the collection of triples $(i, j, A)$ in distributed representation. More precisely, each triple $(i, j, A)$ is encoded as:

$$\mathbf{P}_{left}[i, j, A] \quad = \quad [i]^{\ominus}[j]^{\ominus}[A]^{\ominus} ,$$
$$\mathbf{P}_{right}[i, j, A] \quad = \quad [A]^{\oplus}[i]^{\oplus}[j]^{\oplus} .$$

Then, matrix $\mathbf{P}_{left}$ is the sum of all elements $\mathbf{P}_{left}[i, j, A]$, encoding the collection of all triples from $P$. Similarly, $\mathbf{P}_{right}$ is the sum of all elements $\mathbf{P}_{right}[i, j, A]$. To visualize this representation, the matrix $\mathbf{P}_{left}$

of our running example is represented in the Tetris-like notation in Figure 1, where we used the pieces in Figure 2.

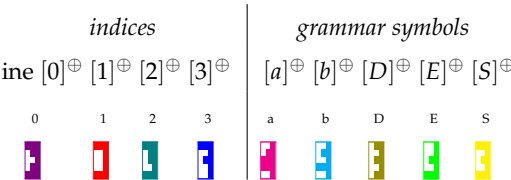

**Figure 2.** Tetris-like graphical representation for the pieces for symbols in our running example.

### 4.2. Encoding and Using Unary Rules

The CYK algorithm uses symbols $a_i$ from the input string $w = a_1 a_2 \cdots a_n$ and unary rules to fill in cells $P[i-1, i]$, as seen in Algorithm 1. We simulate this step in our D-CYK using our distributed representation and matrix operations.

D-CYK represents each input token $a_i$ using a dedicated matrix $\mathbf{P}_w$, defined as:

$$\mathbf{P}_w \quad = \quad \sum_{i=1}^{n} [i-1]^{\ominus}[i]^{\ominus}[a_i]^{\ominus} .$$

For processing unary rules, we use the first part of the D-CYK algorithm, called D-CYK_unary and reported in Algorithm 2. D-CYK_unary takes $\mathbf{P}_w$ and produces matrices $\mathbf{P}_{left}$ and $\mathbf{P}_{right}$ encoding nonterminal symbols resulting from the application of unary rules.

---

**Algorithm 2** D-CYK_unary(string $w = a_1 a_2 \cdots a_n$, matrices $\mathbf{R}_u[A]$) **return** $\mathbf{P}_{left}$ and $\mathbf{P}_{right}$

---

1: $\mathbf{P}_w \leftarrow \sum_{i=1}^{n} [i-1]^{\ominus}[i]^{\ominus}[a_i]^{\ominus}$; $\mathbf{P}_{left} \leftarrow \mathbf{0}$; $\mathbf{P}_{right} \leftarrow \mathbf{0}$;
2: **for** $i \leftarrow 1$ to $n$ **do**
3:      **for** $A \in NT$ **do**
4:          $P_A \leftarrow \sigma(\mathbf{R}_u[A][i]^{\oplus}[i-1]^{\oplus}\mathbf{P}_w)$
5:          $\mathbf{P}_{left} \leftarrow \mathbf{P}_{left} + [i-1]^{\ominus}[i]^{\ominus}[A]^{\ominus}P_A$
6:          $\mathbf{P}_{right} \leftarrow \mathbf{P}_{right} + [A]^{\oplus}[i-1]^{\oplus}[i]^{\oplus}P_A$
7:      **end for**
8: **end for**

---

In D-CYK_unary, we also use matrices $\mathbf{R}_u[A]$, for each $A \in NT$ in the left-hand side of some unary rule. These matrices are conceived of to detect the applicability to matrix $\mathbf{P}_w$ of the rules of the form $A \to a$, where $a$ is some alphabet symbol. Matrix $\mathbf{R}_u[A]$ is defined as:

$$\mathbf{R}_u[A] \quad = \quad \sum_{(A \to a) \in R} [a]^{\oplus} , \tag{1}$$

where $R$ is the set of rules of the grammar. The operation involving $\mathbf{R}_u[A]$ and $\mathbf{P}_w$, which detects whether some rule $A \to a$ is applicable at position $(i-1, i)$ of the input string, is (Line 4 in Algorithm 2):

$$P_A \quad \leftarrow \quad \sigma(\mathbf{R}_u[A][i]^{\oplus}[i-1]^{\oplus}\mathbf{P}_w) ,$$

where $\sigma(x)$ is a sigmoid function $\sigma(x) = \frac{1}{1+e^{-(x-0.5)*\beta}}$. In fact, $[i]^{\oplus}[i-1]^{\oplus}\mathbf{P}_{left} \approx [a_i]^{\ominus}$ extracts the distributed representation of terminal symbol $a_i$. Then:

$$\mathbf{R}_u[A][a_i]^{\ominus} \quad = \quad \sum_{(A \to a) \in R} [a]^{\oplus}[a_i]^{\ominus} \approx \begin{cases} \mathbf{0} & \text{if } (A \to a_i) \notin R \\ \mathbf{I} & \text{if } (A \to a_i) \in R \end{cases} \tag{2}$$

is reinforced by the subsequent use of the sigmoid function. Hence, if some unary rule with left-hand side symbol $A$ is applicable, the resulting matrix is approximately the identity matrix $\mathbf{I}$; otherwise, the resulting matrix is approximately the zero matrix $\mathbf{0}$. Then, the operations at Lines 5 and 6:

$$\mathbf{P}_{left} \leftarrow \mathbf{P}_{left} + [i-1]^{\ominus}[i]^{\ominus}[A]^{\ominus}P_A$$
$$\mathbf{P}_{right} \leftarrow \mathbf{P}_{right} + [A]^{\oplus}[i-1]^{\oplus}[i]^{\oplus}P_A$$

add a non-zero matrix to both $\mathbf{P}_{left}$ and $\mathbf{P}_{right}$ only if unary rules for $A$ are matched by matrix $\mathbf{P}_w$ encoding the input string $w$.

We describe the application of D-CYK_unary using the running example in Figure 1 and the Tetris-like representation. The two unary rules $D \rightarrow a$ and $E \rightarrow b$ are represented as $\mathbf{R}_u[D] = [a]^{\oplus}$ and $\mathbf{R}_u[E] = [b]^{\oplus}$, that is:

$$\mathbf{R}_u[D] = \boxed{\begin{smallmatrix}a\end{smallmatrix}} \qquad \mathbf{R}_u[E] = \boxed{\begin{smallmatrix}b\end{smallmatrix}}$$

in the Tetris-like form. Given the input string $w = aab$, the matrix $\mathbf{P}_w$ is:

$$\mathbf{P}_w = \boxed{\begin{smallmatrix}0 & 1 & a & & 1 & 2 & a & & 2 & 3 & b\end{smallmatrix}}$$

We focus on the application of rule $D \rightarrow a$ to cell $P[0,1]$ of the parsing table, represented through matrices $\mathbf{R}_u[D]$ and $\mathbf{P}_w$, respectively. At Steps 4 and 5 of Algorithm 2, taken together, we have:

$$\mathbf{P}_{left} \leftarrow \mathbf{P}_{left} + \boxed{\begin{smallmatrix}0 & 1 & D\end{smallmatrix}}\,\sigma(\boxed{\begin{smallmatrix}a & & 1 & 0\end{smallmatrix}}\,\mathbf{P}_w) = \mathbf{P}_{left} + Update$$

The *Update* part of the assignment can be expressed as:

$$Update = \boxed{\begin{smallmatrix}0 & 1 & D\end{smallmatrix}}\,\sigma(\boxed{\begin{smallmatrix}a & & 1 & 0\end{smallmatrix}}\,\boxed{\begin{smallmatrix}0 & 1 & a & & 1 & 2 & a & & 2 & 3 & b\end{smallmatrix}})$$

$$\approx \boxed{\begin{smallmatrix}0 & 1 & D\end{smallmatrix}}\,\sigma(\boxed{\begin{smallmatrix}a & & 1 & 1 & a\end{smallmatrix}}) \approx \boxed{\begin{smallmatrix}0 & 1 & D\end{smallmatrix}}\,\sigma(\boxed{\begin{smallmatrix}a & a\end{smallmatrix}}) \approx \boxed{\begin{smallmatrix}0 & 1 & D\end{smallmatrix}}$$

This results in the insertion of the distributed representation of triple $(0, 1, D)$ in $\mathbf{P}_{left}$. A symmetrical operation is carried out to update $\mathbf{P}_{right}$. After the application of matrices $\mathbf{R}_u[D]$ and $\mathbf{R}_u[E]$ at each cell $P[i-1, i]$, the matrices $\mathbf{P}_{left}$ and $\mathbf{P}_{right}$ provide the following values:

$$\mathbf{P}_{left} = \boxed{\begin{smallmatrix}0 & 1 & D & & 1 & 2 & D & & 2 & 3 & E\end{smallmatrix}} \tag{3}$$

$$\mathbf{P}_{right} = \boxed{\begin{smallmatrix}D & 0 & 1 & & D & 1 & 2 & & E & 2 & 3\end{smallmatrix}} \tag{4}$$

### 4.3. Encoding and Using Binary Rules

To complete the specification of algorithm D-CYK, we describe here how to encode binary rules in such a way that these rules can fire over the distributed representation of table $P$ through operations in our matrix algebra. We introduce the second part of the algorithm, called D-CYK_binary and specified in Algorithm 3, and we clarify why both $\mathbf{P}_{left}$ and $\mathbf{P}_{right}$, introduced in Section 4.1, are needed.

---

**Algorithm 3** D-CYK_binary(**P**, rules $R_A$ for each $A$) **return** $\mathbf{P}_{left}$

---

1:　**for** $j \leftarrow 2$ to $n$ **do**
2:　　**for** $i \leftarrow j - 2$ to 0 **do**
3:　　　**for** $A \in NT$ **do**
4:　　　　$P_A \leftarrow \sigma([j]^{\ominus}[i]^{\oplus}\mathbf{P}_{left}\mathbf{R}_b[A]\mathbf{P}_{right}) \otimes \mathbf{I}$
5:　　　　$\mathbf{P}_{left} \leftarrow \mathbf{P}_{left} + [i]^{\ominus}[j]^{\ominus}[A]^{\ominus}P_A$
6:　　　　$\mathbf{P}_{right} \leftarrow \mathbf{P}_{right} + [A]^{\oplus}[i]^{\oplus}[j]^{\oplus}P_A$
7:　　　**end for**
8:　　**end for**
9:　**end for**

---

Binary rules in $R$ with nonterminal symbol $A$ in the left-hand side are all encoded in a matrix $\mathbf{R}_b[A]$ in $\mathbb{R}^{d \times d}$. $\mathbf{R}_b[A]$ is conceived of for defining matrix operations that detect if rules of the form $A \rightarrow BC$, for some $B$ and $C$, fire in position $(i, j)$, given $\mathbf{P}_{left}$ and $\mathbf{P}_{right}$. This operation results in a nearly identity matrix $\mathbf{I}$ for a specific position $(i, j)$ if at least one rule coded in $\mathbf{R}_b[A]$ fires in position $(i, j)$ over positions $(i, k)$ and $(k, j)$, for any value of $k$. This in turn will enable the insertion of new symbols in $\mathbf{P}_{left}$ and $\mathbf{P}_{right}$, for later steps.

To define $\mathbf{R}_b[A]$, we encode the right-hand side of each binary rule $A \rightarrow BC$ in $R$ as $[B]^{\oplus}[C]^{\ominus}$. All the right-hand sides of binary rules with symbol $A$ in the left-hand side are then collected in matrix $\mathbf{R}_b[A]$:

$$\mathbf{R}_b[A] \quad = \quad \sum_{A \rightarrow BC} [B]^{\oplus}[C]^{\ominus}. \tag{5}$$

Algorithm 3 uses these rules to determine whether a symbol $A$ can fire in a position $(i, j)$ for any $k$. The key part is Line 4 of Algorithm 3:

$$P_A \leftarrow \sigma([j]^{\ominus}[i]^{\oplus}\mathbf{P}_{left}\mathbf{R}_b[A]\mathbf{P}_{right}) \otimes \mathbf{I} \,.$$

Matrix $\mathbf{R}_b[A]$ selects elements in $\mathbf{P}_{left}$ and $\mathbf{P}_{right}$ according to rules for $A$. Matrices $\mathbf{P}_{left}$ and $\mathbf{P}_{right}$ have been designed in such a way that, after the nonterminal symbols in the selected elements have been annihilated, the associated spans $(i, k)$ and $(k, j)$ merge into span $(i, j)$. Finally, the terms $[j]^{\ominus}[i]^{\oplus}$ are meant to check whether the span $(i, j)$ has survived. If this is the case, the resulting matrix will be very close to $n\mathbf{I}$, with $n$ depending on the number of merges that have produced the span $(i, j)$; otherwise, the resulting matrix will be very close to the null matrix $\mathbf{0}$. To remove from the resulting matrix the potential $n$ factor, we use the sigmoid function $\sigma(x)$. Finally, we apply an element-wise multiplication with $\mathbf{I}$, which is helpful in removing noise.

Albeit that it is not the focus of our study, it is important to observe the computational complexity of the core part of D-CYK since it was reduced to $O(n^2|NT|)$ where $n$ is the length of the input and $|NT|$ is the cardinality of the set of non-terminal symbols. The computational complexity of the CKY in Algorithm 1 is $O(n^3|R|)$ where $|R|$ is the size of the grammar. The reduction in complexity is justified by the fact that we regard as a constant the parameter $d$ associated with the distributed representation, and by the fact that the inner loop in Algorithm 1 starting in Line 8 is absorbed by the matrix operations in Lines 4-6 of Algorithm 3. Yet, this reduction in complexity is paid with the approximation of the results introduced by these matrix operations over the distributed representation of the CYK matrix.

To visualize the behavior of the algorithm D-CYK_binary and the effect of using $\mathbf{R}_b[A]$, we use again the running example of Figure 1. The only nonterminal with binary rules is $S$, and matrix $\mathbf{R}_b[S]$ is:

$$\mathbf{R}_b[S] = \boxed{\begin{array}{cc} \overset{D\ E}{\blacksquare\blacksquare} & \overset{D\ S}{\blacksquare\blacksquare} \end{array}}$$

Let us focus on the position $(1,3)$. The operation is the following:

$$P_A = \sigma\left( \begin{array}{c} \text{\tiny 3 1} \\ \blacksquare \end{array} \mathbf{P}_{left} \begin{array}{c} \text{\tiny D E} \quad \text{\tiny D S} \\ \blacksquare \quad \blacksquare \end{array} \mathbf{P}_{right} \right) \otimes \mathbf{I}$$

where $\mathbf{P}_{left}$ and $\mathbf{P}_{right}$ are as in Equations (3) and (4). We can then write:

$$\begin{array}{c} \text{\tiny 3 1} \\ \blacksquare \end{array} \mathbf{P}_{left} \begin{array}{c} \text{\tiny D E} \quad \text{\tiny D S} \\ \blacksquare \quad \blacksquare \end{array} \mathbf{P}_{right} =$$

$$\approx$$

$$\approx$$

$$\approx \mathbf{I}$$

Then, this operation detects that $S$ is active in position $(1,3)$.

With this second part of the algorithm, CYK is reduced to D-CYK, which works with distributed representations and matrix operations compliant with neural networks.

## 5. Interpreting D-CYK as a Recurrent Neural Network

Recurrent neural networks (RNNs) are used in the analysis of unbounded length sequences. RNNs exploit the notion of internal state and can be viewed as a special kind of dynamic, non-linear system. In this section, we assume the reader is familiar with the definition of RNN; for an introduction, see for instance [30], Section 8.4. In what follows, we describe an interpretation of our D-CYK algorithm as an RNN. This will show that our D-CKY opens the possibility of implementing RNN organized according to a CYK algorithm.

### 5.1. Building Recurrent Blocks

We start by considering the core part of the D-CYK algorithm, that is Algorithm 3. Let $w = a_1 a_2 \cdots a_n$, $n \geq 1$, be our input string. In order to simulate the two outermost for-cycles in Algorithm 3, the input to the recurrent neural network is a sequence of pairs:

$$([0]^\oplus, [2]^\oplus), ([1]^\oplus, [3]^\oplus), ([0]^\oplus, [3]^\oplus), ([2]^\oplus, [4]^\oplus), ([1]^\oplus, [4]^\oplus), ([0]^\oplus, [4]^\oplus), \ldots, ([0]^\oplus, [n]^\oplus) .$$

The token $([i]^\oplus, [j]^\oplus)$ at position $t$ in the input sequence is denoted $(x_t^{(1)}, x_t^{(2)})$ in what follows.

Recall from Section 3.2 that, for each index $i$, we have $([i]^\oplus)^\top = [i]^\ominus$. We can implement Algorithm 3 through an RNN as follows. We rename matrix $\mathbf{R}_b[A]$ as $\mathbf{W}^A$. We then define recurrent blocks for each nonterminal symbol $A \in NT$:

$$\mathbf{Y}_t^A = \sigma((x_t^{(2)})^\top x_t^{(1)} \mathbf{L}_t \mathbf{W}^A \mathbf{R}_t) \otimes \mathbf{I} \tag{6}$$

$$\mathbf{L}_{t+1}^A = (x_t^{(1)})^\top (x_t^{(2)})^\top [A]^\ominus \mathbf{Y}_t^A \tag{7}$$

$$\mathbf{R}_{t+1}^A = [A]^\oplus x_t^{(1)} x_t^{(2)} \mathbf{Y}_t^A \tag{8}$$

Blocks for different nonterminals are then combined for the next step:

$$\mathbf{L}_{t+1} \;=\; \mathbf{L}_t + \sum_{A \in NT} \mathbf{L}_{t+1}^A \tag{9}$$

$$\mathbf{R}_{t+1} \;=\; \mathbf{R}_t + \sum_{A \in NT} \mathbf{R}_{t+1}^A \tag{10}$$

Here, $\mathbf{L}_{t+1}$ encodes matrix $\mathbf{P}_{left}$, and $\mathbf{R}_{t+1}$ encodes matrix $\mathbf{P}_{right}$ from Algorithm 3, after the first $t \times |NT|$ iterations of the innermost for-cycle in Algorithm 3. Matrices $\mathbf{L}_0$ and $\mathbf{R}_0$ are taken from the output of Algorithm 2.

Finally, Algorithm 2 can be directly implemented by tagging each token $a_i$ from $w$ using unary rules from $R$ and by subsequently encoding the tag and the associated indices $i$ and $i-1$ in our distributed form.

*5.2. Decoding Phase*

The decoding phase is based on the specification of Algorithm 4 reported in Section 6. Let $\mathbf{L}$ be the matrix obtained from Equation (9) after the entire input sequence has been processed by the network in Section 5.1. We apply a simple, memory-less forward layer:

$$y_t = \sigma(\mathbf{W}_d x_t^{(2)} x_t^{(1)} \mathbf{L})$$

to the sequence:

$$([0]^{\oplus}, [1]^{\oplus}), ([0]^{\oplus}, [2]^{\oplus}), ([0]^{\oplus}, [3]^{\oplus}), \ldots, ([0]^{\oplus}, [n]^{\oplus})([1]^{\oplus}, [2]^{\oplus}), ([1]^{\oplus}, [3]^{\oplus}), \ldots, ([n-1]^{\oplus}, [n]^{\oplus})$$

where $\mathbf{W}_d$ is initialized with rows representing the nonterminal symbols in a distributed way.

## 6. Experiments

The aim of this section is to evaluate the D-CYK algorithm by assessing how close it comes to the traditional CYK algorithm, that is how similar the parsing tables produced by the two algorithms are. For this purpose, we use artificial, ambiguous CFGs. Hence, understanding how good D-CYK is at selecting the best syntactic interpretation in an NLP setting is out of the scope of this study.

*6.1. Experimental Setup*

We experiment with five different CFGs in Chomsky normal form, named $G_i$ with $0 \le i \le 4$. Table 1 reports some elementary statistics about these grammars. Each $G_i$ is obtained from $G_{i-1}$ by adding new rules and new nonterminal symbols, so that the language generated by $G_{i-1}$ is always included in the language generated by $G_i$. More precisely, $G_1$ expands over $G_0$ by adding unary rules only; all remaining grammars $G_i$ expand over $G_{i-1}$ by adding binary rules only. Grammars are directly encoded in $\mathbf{R}_u[A]$ and $\mathbf{R}_b[A]$ as described in Equations (1) and (5). These parameters are not learned, but set. We test each $G_i$ with the same set of 2000 strings obtained by randomly generating strings of various lengths using $G_0$.

**Table 1.** Chomsky normal form grammars used in the experiments. We report the total number of rules and the number of binary rules.

| Grammar | Rules | Binary |
|:---:|:---:|:---:|
| $G_0$ | 8 | 5 |
| $G_1$ | 25 | 5 |
| $G_2$ | 28 | 8 |
| $G_3$ | 34 | 14 |
| $G_4$ | 41 | 21 |

As we want to understand whether D-CYK is able to reproduce the original CYK, we use Algorithm 4 to decode $\mathbf{P}_{left}$ into a set $Dec(\mathbf{P}_{left})$ of triples of the form $(i, j, A)$. This set represents the parsing table computed by algorithm D-CYK, as already described in Section 4.1.

---

**Algorithm 4** $Dec(\mathbf{P}_{left})$ **return** table $P$

---

1: **for** $i \leftarrow 0$ to $n-1$ **do**
2:      **for** $j \leftarrow i+1$ to $n$ **do**
3:          **for** each $A$ in $NT$ **do**
4:              **if** $\sigma([A]^{\oplus}[j]^{\oplus}[i]^{\oplus}\mathbf{P}_{left})_{0,0} > 0.99$ **then**
5:                  add $A$ to $P[i,j]$
6:              **end if**
7:          **end for**
8:      **end for**
9: **end for**

---

Set $Dec(\mathbf{P}_{left})$ and the set of triples representing the parse table $P$ obtained by applying the traditional CYK algorithm on the same grammar and string are then compared by evaluating precision and recall, considering $P$ as the oracle, and combining the two scores in the usual way into the *F*1-measure (*F*1). *F*1 measures how much the decoding of matrix $\mathbf{P}_{left}$ is similar to the original parsing table $P$. We experiment with six different dimensions $d$ for matrix $\mathbf{P}_{left}$: 100, 1000, 2000, 3000, 4000, 5000, and 6000.

*6.2. Results and Discussion*

For grammar $G_0$ and for strings of length $\leq 7$, Figure 3 shows that, as the dimension $d$ of matrix $\mathbf{P}_{left}$ increases, D-CYK can approximate more precisely the information computed by the traditional CYK algorithm, as attested by the increase of measure *F*1. The increase of *F*1 is mainly due to an improvement in precision, while recall is substantially stable.

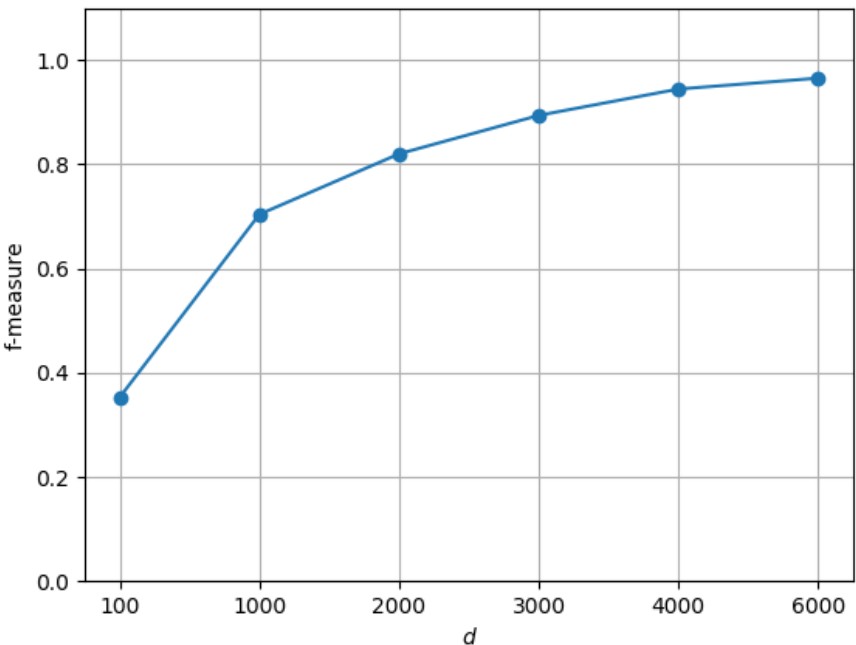

**Figure 3.** *F*1 vs. dimension $d$ on strings with length $\leq 7$ and grammar $G_0$.

On the downside, the length of the input string affects the accuracy of D-CYK. This is seen in Figure 4, reporting several tests on grammar $G_0$ for different dimensions $d$. Even for our highest

dimension $d = 6000$, there is a 20% drop in accuracy, as measured by $F1$, for strings of length eight. The size of the grammar is also a major problem. In fact, the accuracy of the D-CYK algorithm is affected by the number of rules/nonterminals for increasingly larger grammars $G_1$ to $G_4$, as shown in Figure 5.

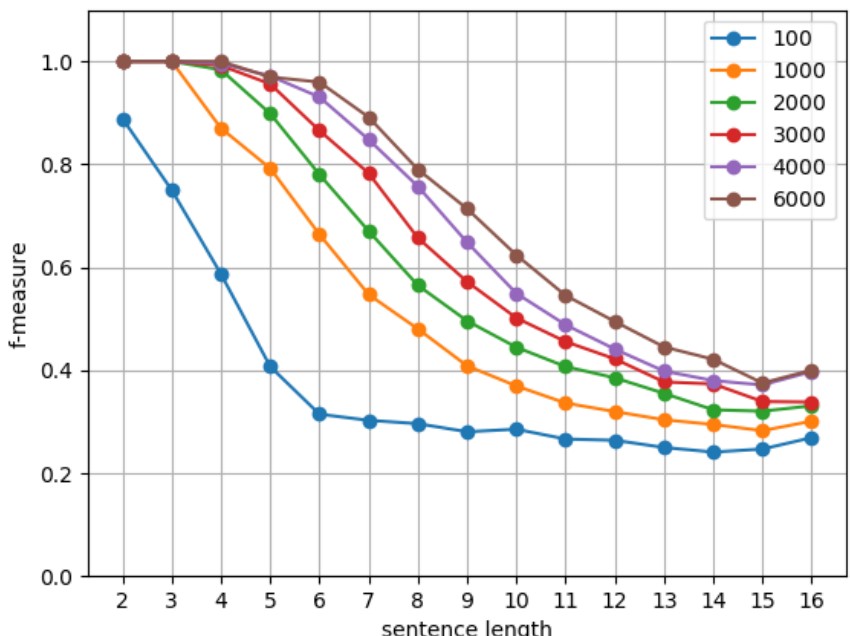

**Figure 4.** $F1$ vs. string length for different dimensions $d$ on grammar $G_0$.

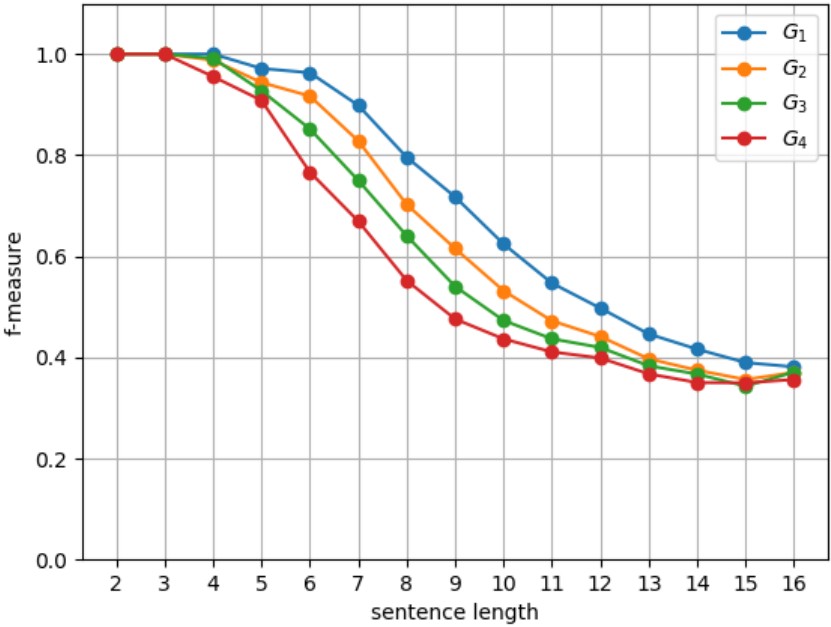

**Figure 5.** $F1$ vs. length of strings with grammars $G_1$, $G_2$, $G_3$ and $G_4$.

In addition to these results, we also experimented with a seq2seqmodel based on LSTM by considering the task as the production of the sequence of symbols in the CKY matrix from the input sequence. We prepared an additional dataset of 2000 examples for training. However, the neural network did not produce significative results. For this reason, we omitted the table here. The task is too complex to be learned by a trivial seq2seq model.

## 7. Conclusions and Future Work

At present, the predominance of symbolic, grammar-based algorithms for parsing of natural language has been successfully challenged by neural networks, which are based on distributed representations. However, Weiss et al. [24] and Suzgun et al. [25] showed the limitations of recurrent neural networks in capturing context-free languages in their full generality, when the input string is provided in plain form. Following this line of investigation, in this short article, we make a first step toward the definition of neural networks that can process context-free languages. We introduce the D-CYK algorithm, which is a distributed version of CYK, a classical parsing algorithm for context-free languages. We also show how to implement D-CYK by means of a recurrent neural network, based on the unfolding of the input string into tokens that encode the search space of all possible constituents. Preliminary experiments show that, to some extent, D-CYK can simulate CYK in the new setting of distributed representations.

In this work, we exploited holographic reduced representations, a specific distributed representation. In future work, we intend to explore alternative distributed representations, possibly using training, for use with our D-CYK algorithm. Neural networks are a tremendous opportunity to develop novel solutions for known tasks. Our proposal opens the avenue of revitalizing symbolic parsing methods in the framework of neural networks for NLP and for other areas working with parsing algorithms and grammars along with neural networks [31,32]. In fact, D-CYK is the first step towards the definition of a "completely distributed CYK algorithm" that builds trees in distributed representations during the computation, and it fosters the definition of recurrent layers of CYK-informed neural networks.

**Author Contributions:** Conceptualization, F.M.Z.; formal analysis, G.S.; software, G.C.; writing—original draft, F.M.Z. and G.S. All authors have read and agreed to the published version of the manuscript.

**Funding:** This research received no external funding.

**Conflicts of Interest:** The authors declare no conflict of interest.

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
