# Peer review of "CYK Parsing over Distributed Representations"

_algorithms, doi:10.3390/a13100262_

Round 1

Reviewer 1 Report

In this manuscript, the authors propose a parsing algorithm called D-CYK, which is a distributed version of the CYK algorithm. It is claimed that the proposed D-CYK can works solely on the basis of distributed representations and matrix multiplication, without any auxiliary symbolic data structures. In my opinion, the manuscript is well written, and the proposed method is reasonable. I only have the following comments.

  1. There is a lack of recent references, especially research works in the past three years.
  2. To increase the completeness of the manuscript, I suggest that the authors add a brief introduction of the Tetris-like graphical representation.
  3. Page 5, Section 5.2, are the words in red color something the authors forgot to remove? I think Wd may be a parameter that should be learned in a training process.
  4. I suggest the authors adding a extra figure to illustrate the RNN they used. What is the structure of the RNN, what are the inputs, and what are the outputs.
  5. There should be a time complexity analysis of the proposed D-CYK.
  6. The performance of the proposed D-CYK should be compared with those of the state-of-the-art paring algorithms.
  7. The RNN is a kind of deep learning approach. Therefore, there should be training data used to learn the parameters of the network. What is the quantity of the training data that the authors used in their experiments? The authors only mentioned that they use 2000 strings in the experiments.

Author Response

We thank the Editor and the reviewers for their thorough reading of our article and their detailed suggestions for how it can be further improved.  In what follows, we respond specifically to each of the points in the decision letter and the reviews and outline the changes that we have made to the article in the revised version.

In the following, your review is in italicus.

Reviewer 1

==========

In this manuscript, the authors propose a parsing algorithm called D-CYK, which is a distributed version of the CYK algorithm. It is claimed that the proposed D-CYK can works solely on the basis of distributed representations and matrix multiplication, without any auxiliary symbolic data structures. In my opinion, the manuscript is well written, and the proposed method is reasonable. I only have the following comments.

1 There is a lack of recent references, especially research works in the past three years.

ANSWER: The goal of our research is modelling the CYK algorithm using distributed representations and matrix multiplication operations. The majority of recent approaches in NLP are disregarding syntactic parsing by means of traditional tabular methods; therefore there is not much to cite from the last three years. 

2 To increase the completeness of the manuscript, I suggest that the authors add a brief introduction of the Tetris-like graphical representation.

ACTION: We have introduced a paragraph describing the Tetris-like graphical representation.

3 Page 5, Section 5.2, are the words in red color something the authors forgot to remove? I think Wd may be a parameter that should be learned in a training process.

ANSWER: Yes, indeed! Sorry for this. We have removed the additional words.

4 I suggest the authors adding a extra figure to illustrate the RNN they used. What is the structure of the RNN, what are the inputs, and what are the outputs.

ANSWER: The focus of our research is to show that the CYK parsing algorithm, which is a symbolic algorithm, can be turned into a distributed algorithm by encoding the CYK matrix into a distributed representation and by performing CYK operations on the basis of matrix multiplication and classical functions that can be used in neural networks.  On the side, and as an example, in Section 5 we also shown that our distributed algorithm can be mathematically expressed as an RNN, opening the way to further investigations which we do not pursue here and we leave for future research.   Hence, in our opinion, in Section 5 we should not put too much emphasis on RNN, as for instance by adding an explanatory figure for the RNN architecture: otherwise the reader may think that this is the focus of our research.

ACTION: We have clarified the above point at the beginning of Section 5, and we have also changed the title of the section 5 accordingly. 

5 There should be a time complexity analysis of the proposed D-CYK.

ACTION: Albeit this is not the main focus of our paper, we have added in section 4.3 a discussion of the time complexity of the core part of our algorithm.

6 The performance of the proposed D-CYK should be compared with those of the state-of-the-art paring algorithms.

ANSWER:  In this research we are aiming to fully reproduce the content of the CKY table using only operations and representations that are compatible with those used by neural networks.  We are therefore dealing with the task of mapping an input string and CFG into a table representing all possible trees.  In contrast, in NLP applications and in other discriminative parsing approaches, parsing is defined as the task of choosing the most likely parse tree, based on some weighted/probabilistic CFG.  The two tasks are different, and direct comparison of the performance for these two tasks does not seem possible to us. 

ACTION: We have clarified this part of the scope of the experiments at the beginning of the experimental section.

7 The RNN is a kind of deep learning approach. Therefore, there should be training data used to learn the parameters of the network. What is the quantity of the training data that the authors used in their experiments? The authors only mentioned that they use 2000 strings in the experiments.

ANSWER: In this research we are considering the task of taking as input a string and a CFG, and of producing as output the associated CYK matrix, doing so entirely on the basis of operations and representations that are compatible with modern neural networks such as RNN.  But the RNN is not used here to learn a grammar model, the grammar is given as input and is directly encoded into the arrays R_u[A] and R_b[A], no learning process is taking place.

ACTION: We have clarified this point in the experimental section.

Reviewer 2 Report

In this study, the authors an algorithm namely D-CYK for parsing over distributed representations. Since the idea is of interest and it might fit to the journal's scope, there are some following issues that need to be addressed:

Major comments:

1. The authors have used NLP and RNN without detail explanation as well as the strength of this combination. A question is raised that if the authors used different NLP models or neural networks, did the performance outperform the D-CYK?

2. The authors convinced that the D-CYK worked well, but they did not provide any comparison to the previous works on the same problem. Without this, it is not easy to ensure that this algorithm outperformed the others.

3. How did the authors select the optimal hyperparameters for deep neural networks?

4. The authors should compare the models with some baseline models to show the significance of the methods.

5. Source codes have to be released for reproducing the results.

Minor comments:

1. There are some grammatical errors. The authors should re-check and revise carefully.

2. The combination of NLP and deep learning has been used in previously published works such as PMID: 31750297 and PMID: 32613242. Therefore, the authors are suggested to refer more works to attract broader readership in this section.

3. Equations should be assigned their numbers. Now the authors missed to assign a lot of equations.

4. Line 316, what is "not clear what Wd is ... where ..."? It is confused here.

5. The authors should have a section title "References" before reference list.

6. More discussions should be added.

Author Response

We thank the Editor and the reviewers for their thorough reading of our article and their detailed suggestions for how it can be further improved. In what follows, we respond specifically to each of the points in the decision letter and the reviews and outline the changes that we have made to the article in the revised version.

In the following, your review is in italicus.

Reviewer 2
==========

In this study, the authors an algorithm namely D-CYK for parsing over distributed representations. Since the idea is of interest and it might fit to the journal's scope, there are some following issues that need to be addressed:

Major comments:

1. The authors have used NLP and RNN without detail explanation as well as the strength of this combination. A question is raised that if the authors used different NLP models or neural networks, did the performance outperform the D-CYK?

ANSWER: We have considered the task at end, namely the translation of an input string to its associated CYK matrix, as a seq2seq problem, and have implemented an LSTM for its solution. Yet, the LSTM was not able to learn how to map input sequences to CYK matrices.

ACTION: We have added to the final discussion section a short sentence describing this pilot experiment.

2. The authors convinced that the D-CYK worked well, but they did not provide any comparison to the previous works on the same problem. Without this, it is not easy to ensure that this algorithm outperformed the others.

ANSWER: In this research we are setting up a new problem: given as input a string and a CFG, produce the CYK matrix associated with the sentence with respect to the grammar by using only operations and representations that are compatible with modern neural networks. This task is new, and there is no relevant work in current and past research moving toward this direction. The literature in NLP is mainly concerned with the task of selecting the best or the N-best parse trees for an input string, based on some weighted/probabilistic CFG. This second task does not take the grammar as input, it rather learns the grammar from a very large amount of training data. In contrast, in our task the grammar is given as input and is directly encoded into the arrays R_u[A] and R_b[A], no training process is taking place in our task. The two tasks are rather different, we cannot directly compare our model with current parsers or current neural networks realizing the parsing task in NLP. We are working on realizing an approach that can exactly replicate CYK parsing in its original, symbolic formulation. This is why we are only comparing our model to the original CYK model, in different configurations.

ACTION: We have clarified this point in the experimental section.

3. How did the authors select the optimal hyperparameters for deep neural networks?

ANSWER: The paper aims to demonstrate that CKY can be implemented over a distributed representation of the CYK matrix and that this implementation enables the possibility of designing a specific neural network. Hence, we illustrate in Section 5 that the algorithm D-CYK can be seen as a RNN. We are not learning the RNN in this study. The RNN intepretation is not actually implemented and, then, learning is not taking place. Grammar is directly encoded in R_u[A] and R_b[A] and not learned.

ACTION: We have clarified this at the beginning of Section 5 and by changing the title of this section, and we have clarified this also in the experimental section.

4. The authors should compare the models with some baseline models to show the significance of the methods.

ANSWER: The baseline is too trivial. We have compared different versions of the algorithm by changing the dimension of the encoding.

5. Source codes have to be released for reproducing the results.

ANSWER: We will release the code in github

Minor comments:

1. There are some grammatical errors. The authors should re-check and revise carefully.

ANSWER: We have re-checked and corrected some typos. But if you have spotted some additional typos, please let us know!

2. The combination of NLP and deep learning has been used in previously published works such as PMID: 31750297 and PMID: 32613242. Therefore, the authors are suggested to refer more works to attract broader readership in this section.

ANSWER: Thank you for the suggestion. Broadening the audience is extremely important.

ACTION: We have added the suggested work and we have added a sentence in the conclusion and future work section.

3. Equations should be assigned their numbers. Now the authors missed to assign a lot of equations.

ANSWER: We can do this if it is really necessary. In the current version, we have assigned numbers only to those equations which are referred to in the text.

4. Line 316, what is "not clear what Wd is ... where ..."? It is confused here.

ANSWER: Yes, indeed! Sorry for this. We have removed the additional words.

5. The authors should have a section title "References" before reference list.

ANSWER: We are using the recommended journal style files, we do not understand the source of this problem ... we suppose this will be solved in the camera ready if the paper is accepted.

6. More discussions should be added.

ACTION: We have added some discussion.

Round 2

Reviewer 2 Report

My previous comments have been addressed well.